# Protein Expression Correlates Linearly with mRNA Dose over Up to Five Orders of Magnitude In Vitro and In Vivo

**DOI:** 10.3390/biomedicines9050511

**Published:** 2021-05-05

**Authors:** Alexander H. van Asbeck, Jürgen Dieker, Rik Oude Egberink, Lennard van den Berg, Johan van der Vlag, Roland Brock

**Affiliations:** 1Department of Biochemistry, Radboud Institute for Molecular Life Sciences (RIMLS), Radboud University Medical Center, 6525 GA Nijmegen, The Netherlands; vanasbeck@ribopro.eu (A.H.v.A.); dieker@mercurna.com (J.D.); rik.oudeegberink@radboudumc.nl (R.O.E.); lennard.vandenberg@radboudumc.nl (L.v.d.B.); 2Department of Nephrology, Radboud Institute for Molecular Life Sciences (RIMLS), Radboud University Medical Center, 6525 GA Nijmegen, The Netherlands; johan.vandervlag@radboudumc.nl; 3Department of Medical Biochemistry, College of Medicine and Medical Science, Arabian Gulf University, Manama 293, Kingdom of Bahrain

**Keywords:** messenger RNA, dose response function, nanomedicine, drug delivery

## Abstract

Messenger RNA is rapidly gaining significance as a therapeutic modality. Here, we address the dependence of dose–response functions on the type of delivery vehicle, cell line, and incubation time. Knowledge of these characteristics is crucial for the application of mRNA. As delivery vehicles, a lipid-based formulation and the cell-penetrating peptide Pepfect14 (PF14) were employed. As cell lines, we included a glomerular endothelial cell line (mGEnC) as a model for differentiated cells, HeLa cells, and SKOV-3 ovarian carcinoma cells. Uptake and expression were detected by flow cytometry, using a Cy5-labelled mRNA coding for enhanced green fluorescent protein (EGFP). There was a linear correlation of dose, uptake, and expression, and this correlation was maintained for over up to 72 h. Through application of a multistep kinetic model, we show that differences in expression levels can already be explained by the number of mRNAs packaged per delivery vehicle. Using luciferase as a reporter protein, linearity of expression was observed over 5 orders of magnitude in vitro and 3 orders of magnitude in vivo. Overall, the results demonstrate that mRNA provides excellent quantitative control over protein expression, also over extended periods of time.

## 1. Introduction

The delivery of messenger RNA as a therapeutic modality has just had its break-through in the clinical approval of two SARS-CoV2 vaccines [1,2]. Next to preventive vaccination, applications range from therapeutic vaccination to protein replacement therapy [3]. In comparison with standard gene therapy, mRNA provides several advantages: There is no possibility of genomic insertion, thereby eliminating the risk of malignant transformation of cells. In addition, exogenous mRNA-mediated protein expression is only transient, which provides temporal control of the therapeutic intervention. Furthermore, mRNA also works for non-dividing cells as there is no requirement for nuclear entry, and for a cell population, the onset of protein expression is more synchronized than for plasmid DNA [4].

As potential disadvantages, mRNA is more prone to degradation than DNA and may activate responses of innate immunity that range from induction of inflammatory reactions to a shut-down of protein expression in target cells [5]. Nucleoside modifications and cap analogs have been developed to prevent innate immunity [6,7,8]. However, unmodified mRNA also yields effective protein production in vitro and in vivo [9].

As with any oligonucleotide, naked mRNA enters cells only inefficiently, even though exceptions seem to exist for some applications, including intramuscular application [10]. To this point, most delivery agents for mRNA are lipid-based. One of the most popular methods for transfection in tissue culture experiments uses aqueous solutions of micellar cationic lipids mixed with mRNA, so-called lipofection [11]. For in vivo delivery, lipid nanoparticles have shown high efficacy and are leading the way to the clinic [3]. As an alternative approach, transfection-competent nanoparticles can be generated through electrostatic complexation with cationic cell-penetrating peptides. One peptide that has been shown to yield effective cellular delivery for a variety of oligonucleotides in vitro and in vivo is PepFect 14 (PF14) [12,13]. Recently, we employed PF14 to deliver mRNA in several cellular models of ovarian cancer and also in a peritoneal model of ovarian cancer in vivo in mice [14].

PF14 belongs to the class of amphipathic CPPs [15]. A commonality of oligonucleotide-CPP nanoparticles is that cellular uptake occurs through induction of endocytosis. Following uptake, it is generally acknowledged that efficient endosomal release is one of the main characteristics distinguishing an active from an inactive delivery agent [16,17,18].

As a consequence, the dose–response relationship of mRNA exposure and protein expression is a function of uptake, endosomal release, mRNA decomplexation and translation, and mRNA degradation at each of these steps. Ideally, for any type of intervention there should be a linear relationship between dose and function. However, the extent to which protein expression follows a linear dose dependence, given the multitude of involved processes, is all but clear.

Here, we employed Cy5-labelled mRNA coding for the reporter protein enhanced green fluorescent protein (EGFP) to investigate dose–response functions on the level of the total cell population and on individual cells. A lipid-based transfection agent and PF14 were employed for delivery using glomerular endothelial cells (mGEnC) [19], SKOV-3 ovarian cancer cells [20], and HeLa cells as cellular models. mGEnC are conditionally immortalized cells that can be differentiated into a non-dividing endothelial phenotype, and we used these cells for mRNA delivery in the differentiated state.

For lipofectamine, there was a linear correlation of mRNA uptake and protein expression over the whole concentration range, whereas for PF14 only above a concentration threshold was a linear increase observed. Importantly, the linearity was maintained over up to 72 h, demonstrating that mRNA yields excellent quantitative control over protein expression. Using luciferase mRNA, we demonstrate that this linear dependence of dose and expression extends over up to 5 orders of magnitude and also to expression in vivo.

A multistep kinetic model of mRNA delivery could recapitulate several key observations—namely, that even assuming equal probabilities for endosomal release, the simulations for lipofectamine yielded higher average EGFP expression than for PF14 [21]. This higher expression is a consequence of the fact that per lipofectamine particle more mRNAs are imported.

## 2. Materials and Methods

### 2.1. Messenger RNA

Cy5-labelled EGFP mRNA, modified with 5-methoxyuridine, capped using CleanCap technology, and polyadenylated were purchased from Trilink Biotechnologies (San Diego, CA, USA). The length of the mRNA was 996 nucleotides, and Cy5-EGFP mRNA contained a 3:1 methoxyuridine ratio with Cy5. NanoLuc mRNA was obtained by in vitro transcription using the T7 HighScribe kit (ThermoFisherScientific, Waltham, MA, USA) according to manufacturer’s specifications, with 200 ng of purified PCR product (F primer: AATTAATACGACTCACTATAGGGATACGCCGCCACCATGAACTCCTTCTCCACAAGC, R primer: GTATCTTATCATGTCTGCTCGAAG, Q5-high fidelity polymerase (NEB, Ipswich, MA, USA), purified with MinElute PCR purification kit (Qiagen Benelux, Venlo, The Netherlands)) as input. NanoLuc mRNA was capped with Vaccinia Capping enzymes (NEB) and extended with a 250 nt poly-A-tail (*E. coli* polyA polymerase kit, NEB) according to manufacturer’s specifications. Final purification of the mRNA was performed using the RNeasy RNA purification kit (Qiagen) according to manufacturer’s specifications, with elution in RNAse-free MQ. All mRNA was stored at 1 µg/µL in MQ at −80 °C until use. Before use, the mRNA solution was thawed at RT and stored on ice. CleanCap-Fluc mRNA was purchased from TriLink Biotechnologies (San Diego, CA, USA).

### 2.2. Tissue Culture

Conditionally immortalized mouse glomerular cells (mGEnC) were proliferated as described previously at 33 °C [19]. The human cervical carcinoma cell line HeLa and the ovarian carcinoma SKOV-3 cell line were obtained from the American Type Culture Collection (ATCC) and were maintained in Dulbecco’s Modified Eagle Medium (Gibco) supplemented with 10% fetal calf serum (PAN-Biotech) and GlutaMAX (Life Technologies, Carlsbad, CA, USA) in a 37 °C, 5% CO_2_ humidified incubator.

### 2.3. Nanoparticle Formation

Lipofectamine MessengerMAX, a formulation specifically optimized for mRNA transfection in vitro (ThermoFisherScientific), was used according to manufacturer’s specifications. Briefly, a multiple of 0.15 µL of Lipofectamine MessengerMAX was mixed with the same multiple of 5 µL of Opti-MEM medium (RT, ThermoFisherScientific) and incubated for 10 min at RT. Subsequently, this medium was mixed by repeated pipetting with a corresponding multiple of 100 ng of mRNA (1 µg/µL), which was immediately before mixing prediluted with the same multiple of 5 µL of Opti-MEM (RT). The resulting solution was incubated for 5 min at RT before subsequent dilution with Opti-MEM medium (RT) to the desired concentration, depending on the desired dose. For all indicated doses, the same volume of 10 µL mRNA formulation was added to 90 µL of medium per well to obtain a total volume of 100 µL medium.

Pepfect14 (Stearyl-AGYLLGKLLOOLAAAALOOLL-NH_2_, where O stands for ornithine and -NH_2_ for a C-terminal amidation) was a kind gift of Dr. Hällbrink (Stockholm University, Stockholm, Sweden). The peptide was dissolved in RNase-free MQ at a concentration of 3 mM and incubated at RT for 20 min before aliquots were frozen at −20 °C. Particle formation with PF14 was performed by a “stream method”. Two separate stock solutions of peptide and mRNA were prepared in MilliQ water and aspirated in two pipette tips that were connected with tubing to two 3 mL syringes placed in a syringe-pump (PHD Ultra, Harvard apparatus, Holliston, MA, USA). The pipette tips were inserted into a custom-made holder in which a 1.5 mL Eppendorf tube was placed to collect the nanoparticle solution. The angle between both pipette tips was 35 degrees, and the angle between the tips and the tube wall was 45 degrees. The pump was set at an output flow rate of 9 mL/min. PF14 nanoparticles were formed by stream mixing at an N/P ratio of 3 and a concentration of 50 µM PF14. In the N/P ratio, N (nitrogen) specifies the number of positive charges and P (phosphate) the number of negative charges. This stock solution of nanoparticles was subsequently diluted with Opti-MEM to obtain the desired final concentration of mRNA, which was directly added to the cells in a volume of 10 µL per well to obtain a final volume of 100 µL. In all cases, final concentrations of PF14 were below 2 µM to avoid toxicity. The hydrodynamic size of the nanoparticles was determined at 25 °C by dynamic light scattering (DLS) using a Zetasizer Nano ZS apparatus (Malvern Instruments, Worcestershire, UK) in MQ with ZEN0040 cuvettes.

### 2.4. Detection of Luciferase Expression

Secreted NanoLuciferase activity was detected with the Nano-Glo Luciferase Assay System (Promega, Madisson, WI, USA) according to manufacturer’s specifications. Importantly, the assay buffer was thawed and equilibrated to RT for more than 1 h at RT.

### 2.5. Confocal Microscopy

mGEnC cells were differentiated for 1 week in µ-slide 8-well chambers (Ibidi, Gräfelfing, Germany) in DMEM:HAMF12 (3:1) medium (Gibco, Gaithersburg, MD, USA) containing 10% FCS (PAN-Biotech, Aidenbach, Germany) at 37 °C to 100% confluency. Freshly prepared nanoparticles were added to the cells by mixing 25 µL of a 10× stock solution of the nanoparticles with 225 µL medium. Cells were incubated for 24 h and washed twice with PBS before the addition of fresh medium. A portion of 50 nM Lysotracker Red (ThermoFisherScientific) was added to the medium and incubated for 30 min under differentiation conditions, which means cultivation at 37 °C. After replacing the lysotracker solution with fresh medium, cells were directly imaged using an SP5 confocal microscope (Leica Microsystems, Mannheim, Germany) equipped with an HCX PL APO 63× NA 1.2 water immersion lens. Cells were maintained during the measurement at 37 °C on a temperature-controlled microscope stage.

### 2.6. Flow Cytometry

For flow cytometry cells of 3 wells of a 96-well plate were combined with about 50,000 HeLa and SKOV-3 cells per well and 10,000 mGEnC per well. mGEnC were differentiated as described above. Cells were incubated for 24 h with freshly prepared nanoparticles containing the indicated concentration of mRNA by adding 25 µL of a 10× stock solution to 225 µL medium. After incubation, cells were washed twice with PBS and detached using Trypsin/EDTA. Cellular fluorescence of approximately 10,000 cells was measured using a FACSCaliber flow cytometer (BD Biosciences). Subsequently, the gated population was analyzed using FlowJo software.

### 2.7. Application of mRNA in Mice

An amount of 10 µg Fluc mRNA was formulated with Lipofectamine MessengerMax as described above and diluted in Opti-MEM to the concentration used for injection, which took place within 30 min after formulation. BALB/cJRj mice (Janvier Labs, Le Genest-Saint-Isle, France) were injected in the tail vein with 50 µL of the solution containing the indicated concentration of formulated mRNA. After 24 h, organs were collected, rinsed with PBS to remove blood, and kept for maximally 1 h in PBS on ice. Organ lysates were prepared by adding a volume of lysis buffer (10 mM Tris HCl pH 7.8; 10% glycerol; 1% Triton X-100; 1% Tween-20; 0.31 mg/mL DTT) containing protease inhibitors (Complete Ultra Tablets, Roche, Basel, Switzerland) that corresponded to 5× the weight of the organ. After homogenization using a TURRAX disperser (IKA, Staufen, Germany), tissues were incubated for 2 h on ice. Supernatants were collected after centrifugation of the lysate for 10 min at 20,800× *g*. Firefly luciferase activity was measured using the Luciferase Assay Reagent (Promega) according to the manufacturer’s protocol.

### 2.8. Computational Simulation of Transfection

Simulations of transfections were based on a stochastic implementation of the multistep mRNA delivery model described by Ligon et al. [21,22]. The uptake and delivery models were simulated using Complex Pathway Simulator (COPASI) 4.28 (Build 226). For stochastic modelling, 100 iterations were used to calculate averages and geometric means. Duration of the time course was set to 120 h with 30 min intervals. PF14 to LipoMM ratio calculations were performed at peak response times for transient values (GFP; t = 21 h, mRNA; t = 6 h) and at the endpoint for cumulative values.

## 3. Results

This section may be divided by subheadings. It should provide a concise and precise description of the experimental results, their interpretation, as well as the experimental conclusions that can be drawn.

### 3.1. Messenger RNA Uptake and Reporter Gene Expression Depend on Cell Line and Delivery Vehicle

Protein expression from exogenously delivered mRNA is a multistep process involving cellular uptake, endosomal release, unpackaging of mRNA, and finally mRNA translation. At each of these steps, degradation of mRNA may also occur [21]. The extent to which protein expression correlates with dose may therefore be affected by the presence of cooperative effects, saturation, or threshold effects at any of these steps. Cooperativity could, for example, occur if accumulation of nanoparticles inside endosomes would promote endosomal release.

As delivery agents, we employed the lipid-based transfection agent lipofectamine messenger max (LipoMM) and the cationic cell-penetrating peptide PF14. Both delivery vectors had shown potency in mRNA delivery before, even though at least in two-dimensional tissue cultures the activity of LipoMM exceeded the one of PF14 by about 1 order of magnitude, and both vectors were used at concentrations of which we had previously determined to not cause any toxicity [14]. As cell lines, HeLa cells, SKOV-3 ovarian cancer cells, and murine glomerular endothelial cells (mGEnC) were employed. HeLa cells are a well-established carcinoma cell line. SKOV-3 were selected as a cancer model with which we had investigated mRNA delivery in vitro and in vivo before [14]. mGEnC are a conditionally immortalized endothelial cell line that can be differentiated into endothelial cells by shifting the tissue culture temperature from 33 °C to 37 °C. These cells serve as an endothelial model for glomerular kidney diseases [19]. In our case, we used the fully differentiated non-dividing cells to investigate mRNA delivery also in a difficult-to-transfect cell type of pathophysiological relevance.

To be able to directly relate uptake with expression on a single cell level, we employed a Cy5-labelled mRNA coding for EGFP. Two-dimensional histograms of uptake and expression revealed remarkable differences between cell lines and delivery agents (Figure 1, Appendix A). As observed before [14], LipoMM yielded more efficient expression than PF14. mGenC showed the lowest uptake and expression, followed by Hela and SKOV-3 cells. Interestingly, for LipoMM, for all three cell lines there was a population of cells that showed neither mRNA uptake nor EGFP expression. However, depending on the cell line, the size of this population depended on the mRNA dose. Moreover, for SKOV-3 cells, at a dose above 60 ng per well, all cells that showed uptake also showed expression. By comparison, for mGEnC and HeLa cells, at a level of mRNA uptake at which expression was observed, some cells did not show expression. Nevertheless, as a remarkable difference from mGEnC, once a HeLa cell showed expression, it was clearly separated from the population that showed uptake but no expression.

By comparison, PF14 on mGEnC led to a homogenous concentration-dependent shift of the cell population to higher fluorescence intensities for mRNA uptake without any EGFP expression. For HeLa cells and SKOV-3 cells, there was a transition from a regime in which an increase in uptake led to only minor differences in EGFP expression to a regime where small increases in uptake led to large differences in expression. Interestingly, for PF14 there was no subpopulation of cells that did not show any uptake.

### 3.2. Linear Correlations of Dose, Uptake, and Expression Are Maintained over Time

After demonstrating that uptake and expression differed for cell lines and delivery vectors, we were interested in investigating in more detail the quantitative dependence of uptake and expression on mRNA dose, how expression correlated with uptake, and in particular how these correlations evolved over time. We selected mGEnC and Hela cells as they had shown pronounced differences in the initial experiment (Figure 1).

For mGEnC, already after 1 h there was a linear correlation of dose and uptake for both LipoMM and PF14 (Figure 2). For LipoMM, this linear correlation of dose and uptake also translated into a linear dependence of dose and expression from 2 h on, and these linear relationships persisted over the whole 72 h time course of the experiment. The same was true for the dependence of uptake on dose for PF14 even though, as observed before, there was no EGFP expression at any time point.

For Hela cells treated with LipoMM-formulated mRNA, there was a linear dependence of uptake and EGFP expression on dose after 4 h (Figure 3). This linearity was then maintained over the whole 72 h time course of the experiment, even though peak uptake and expression at 24 h were about 40 and 100 times as strong as at 4 h and decreased to one-third of the peak values by 72 h. Inspection of the two-dimensional scatter plots revealed that at 4 h, expression emerged from the fraction of cells with the highest uptake, whereas at 24 h the vast majority of cells contributed to expression (Appendix A).

PF14 on the other hand, was more effective in inducing uptake, yielding a clear linear correlation of dose and uptake already after 2 h, and a positive correlation of dose and expression was detected at 8 h, which further evolved into a linear correlation over the remainder of the experiment.

The maintenance of a linear dose–response function over time strongly indicated that the temporal evolution of uptake and expression was independent of dose. To substantiate this independence of time further, we plotted uptake and expression versus time, in dependence of dose (Figure 4). For a given cell type, treatment, and readout (uptake/EGFP expression), all curves showed the same time-course, independent of dose. A notable exception was EGFP expression for PF14-mediated uptake in HeLa cells, for which a concentration threshold had to be crossed to yield EGFP expression.

In addition, there was a stronger decrease in Cy5 fluorescence from 24 to 72 h for the PF14-delivered mRNA than for the LipoMM-delivered mRNA. This difference was even more prominent when the *y* axes were plotted on a linear scale (Appendix A). By comparison, the levels of EGFP expression followed the same kinetics in both mGEnC and HeLa cells with only a slight drop at later time points. The independence of the temporal evolution on dose was also apparent when for the individual concentrations EGFP fluorescence was plotted versus uptake over time (Appendix A). This plot further substantiated the differences between delivery affected by LipoMM and PF14: For PF14 there was a stronger drop in cellular Cy5 fluorescence.

### 3.3. Peptide-Mediated Delivery of mRNA Leads to Stronger Endolysosomal Retention Than Lipid-Mediated Delivery

Considering the striking difference between PF14 and LipoMM in their capacity to induce cellular uptake of mRNA and protein expression, we investigated the subcellular distribution of mRNAs using confocal microscopy. We hypothesized that the lower expression of EGFP in spite of a stronger Cy5 uptake and also the accelerated loss of Cy5 fluorescence in comparison with LipoMM-formulated mRNA, could be explained by a stronger retention of mRNA-PF14 polyplexes in the endolyosomal compartment and degradation of mRNA in this compartment.

In agreement with flow cytometry, the cellular mRNA signal for PF14-treated cells was more intense than the signal for LipoMM transfected cells (Figure 5a). Nevertheless, EGFP expression was only observed for the latter. In neither case was there an indication of lipo-/polyplexes that had remained associated to the cell surface. Instead, Cy5 fluorescence was heterogeneously distributed throughout the cytoplasm. Interestingly, there was a pronounced difference between PF14 polyplexes and LipoMM polyplexes in the degree of colocalization with lysotracker. For PF14, nearly all Cy5-mRNA fluorescence colocalized with lysotracker and vice versa, indicating poor to no endosomal release. By comparison, for LipoMM the mRNA showed only partial colocalization with lysotracker, while lysotracker showed a high degree of colocalization with mRNA, consistent with endosomal release of part of the mRNA into a translation-competent form. These differences in the colocalization were in agreement with our initial hypothesis that the more pronounced loss in Cy5 fluorescence was a consequence of endolysosomal breakdown.

### 3.4. Protein Expression Linearly Correlates with mRNA Dose over Five Orders of Magnitude

After having established a linear correlation of mRNA dose with EGFP expression over a dose range of 1 order of magnitude, we wanted to explore the degree to which we could extend the dose range. For this purpose, we used luciferase as a reporter protein and exposed Hela cells and mGenC to LipoMM and PF14-formulated mRNA.

In all cases, there was a linear correlation of reporter gene expression and mRNA dose (Figure 6). Lipofectamine yielded higher expression levels than PF14 by a factor of about 1000, and in HeLa cells, expression at the same dose was higher than in mGEnC, consistent with the results for EGFP expression. In mGEnC, PF14 formulated mRNA yielded no detectable protein expression. When different dose ranges were tested in subsequent experiments, the data points seamlessly connected to each other. For HeLa cells transfected with lipofectamine, the linear dose dependence extended over nearly 5 orders of magnitude.

### 3.5. Additionally, In Vivo, Reporter Gene Expression Correlates Linearly with mRNA Dose

To explore whether the linear dose–response function observed in cell lines also applied to an in vivo situation, we injected BALB/c mice intravenously with a dose range of firefly luciferase-encoding mRNA formulated with LipoMM. Firefly luciferase can easily be detected in a quantitative way in cell extracts. In preliminary experiments, we had observed that intravenous injection of LipoMM-formulated firefly luciferase mRNA predominantly targeted the spleen, while other organs did not reveal any detectable luciferase activity (Appendix A). This provided the opportunity to study the dose–response curve in a single organ. For doses ranging from 0.001 to 10 µg mRNA, we injected each dose into five mice and determined luciferase activity in spleen lysates after 24 h (Figure 7). Largely in accordance with the in vitro results, a linear correlation was observed for 3 orders of magnitude between 0.001 and 1 µg. Bending of the curve could be observed for lower concentrations, which were close to the detection limit of the assay (~50 RLU) and for higher concentrations. Since mice did not demonstrate a significant decrease in weight after 24 h (Appendix A) or any other obvious sign of disturbance, we do not expect that the latter was due to toxicity.

### 3.6. Computational Simulation of mRNA Uptake and Delivery Demonstrates That Differences in Reporter Gene Expression Can Be Explained by the Number of mRNA Molecules per Nanoparticles Also at the Same mRNA Dose

Finally, we were interested in better understanding the possible molecular basis for the differences in expression levels between PF14- and LipoMM-mediated delivery. To quantitatively capture the number of mRNA molecules at each step of the delivery pathway and the resulting expression of reporter proteins, we made use of a multistep kinetic model (Figure 8). A major difference in PF14- and LipoMM-mediated delivery is the number of mRNA molecules per nanoparticle. For PF14-based nanoparticles, we assumed three mRNA molecules per particle based on a recent study of peptide-mRNA polyplexes by superresolution microscopy [23]. For LipoMM, we followed the estimate of Ligon et al. of about 350 mRNA molecules per nanoparticle and—for ease of scaling—adjusted this to 300 mRNA molecules per particle. At the same input of mRNA molecules, this translated into a 100-fold difference of nanoparticles contacting the cells (Table 1).

Importantly, the simulation employed a stochastic implementation of reaction kinetics. The simulated time courses were very similar to the experimental time courses for HeLa cells, with peak EGFP expression levels at about 20 h (Figure 8). As one interesting observation, we noted that for LipoMM-mediated delivery there was a population of EGFP negative cells, both a 10 and 100 ng. At 10 ng mRNA, this amounted to 37% of the total cell population. Remarkably, even though cells were exposed to the same initial mRNA dose for LipoMM, this resulted in 4.5 times higher average EGFP expression at a dose of 10 ng and 5.6 times higher average EGFP expression at 100 ng of mRNA. A characteristic of the multistep model is the explicit consideration of endosomes containing different numbers of nanoparticles, which means that not every particle contacting a cell will lead to the formation of a distinct endosome. As a consequence, while the number of particles contacting the cell directly scales with mRNA dose, this is not the case for the number of formed endosomes. Moreover, even though for PF14 100 times more particles contact a cell, only 10 times more endosomes are formed. From 10 to 100 ng of mRNA, the number of endosomes for both vectors only scales with a factor of 3 instead of a factor of 10. Importantly, for LipoMM-packaged mRNA, endosomal rupture leads to release of 100 times more mRNA molecules per particle than for PF14-packaged mRNA. Therefore, even with fewer endosomes, more mRNA molecules are released into the cytoplasm, leading to protein expression.

## 4. Discussion

For a panel of three different cell lines and two different delivery vehicles, we here show a linear correlation of protein expression with mRNA dose. This linearity also extended to in vivo expression, where linearity was observed over 3 orders of magnitude. Importantly, this linearity was maintained over time.

Next to these general commonalities, there were some notable differences that could be related to characteristics in cellular uptake and intracellular trafficking. Overall, the lipid-based transfection agent had higher activity than the cell-penetrating peptide PF14.

We had observed before that PF14 exceeded other CPPs in their capacity to deliver oligonucleotides into the cytosol [16,24]. Here, we observed that for all three cell lines, PF14 induced cellular uptake. However, the level of EGFP expression strongly differed between cell lines, with mGEnC showing the least and SKOV-3 cells showing the highest level of expression. Most likely, these differences related to the efficiency in endosomal release. A more detailed comparison of mGEnC and SKOV-3 cells may help to better understand characteristics of the endocytic machinery associated with induction of cytosolic delivery or endosomal capture.

Interestingly, with respect to uptake and expression, PF14 yielded a more homogenous response across the cell population than LipoMM. For LipoMM, for all three cell lines, there was a fraction of cells that did not show any uptake. Moreover, cells that showed the same level of uptake were split into subpopulations that either did or did not show expression.

By comparison, PF14 led to a concentration-dependent shift of the whole cell population. In combination with our simulations, our analyses offer two potential explanations: First, this difference may be due to the fact that PF14, being a cell-penetrating peptide, acts by induction of endocytosis [25]. Secondly, due to the lower number of LipoMM particles, not all cells may be exposed to nanoparticles, and with only few endosomes per cell, there is a higher chance that no endosomal release occurs.

For lipofectamine-dependent mRNA delivery, Leonhardt et al. modelled mRNA delivery and release by a simple two-step process comprising endosomal uptake and endosomal release [4]. This model was sufficient to also explain why a fraction of cells did not show protein expression. Key to the interpretations by Leonhardt et al. was that for the chosen experimental conditions only a small number of polyplexes were delivered to each cell. Based on the number of polyplexes that were deposited on a glass surface, they postulated uptake of only 4–8 polyplexes per cell at an mRNA dose of 1 μg in 1 mL total incubation volume. This assumption is in accordance with time-resolved microscopy of lipoplex delivery [26]. Their model suggests a release probability of 25–50% from an endosome. Once the lipoplexes are inside the endosome, release is stochastic and independent of the number of lipoplexes. These observations are consistent with our findings of a cell population that is negative for uptake and expression.

Additionally, in our case, application of an extended version of the original model [21] and making the simple assumption that LipoMM and PF14 polyplexes differed with respect to the number of incorporated mRNA molecules by a factor of one hundred, was sufficient to explain key differences in delivery and expression for both delivery agents. Incorporating our parameter set of concentrations and cell numbers, due to the lower total number of particles present in the LipoMM samples, for stochastic reasons, not all cells were hit by a particle or showed endosomal release, readily explaining the experimentally observed fraction of Cy5- and EGFP-negative cells. Remarkably, these differences in packaging were also sufficient to explain a difference in cellular expression levels by a factor of 3.5 to 6.5, depending on concentration, even when assuming the same probability for endosomal release. The difference may be explained by the fact that for LipoMM one individual event releases 2 orders of magnitude more mRNA into the cytosol. Nevertheless, in reality, both vectors should also differ in release efficiency. Interestingly, the modelled data differed from the experimental observations in that there was no direct linear correlation with dose. A 10-fold increase in mRNA dose only led to an increase in EGFP expression by a factor of about 5. These differences may already be attributed to differences in the number of liposomes that do not scale with dose. As more vectors hit a cell, more are packed together into one endosome instead of forming new endosomes. For the experimental values, the slope of the linear correlation varied with incubation time.

For PF14, in HeLa and SKOV-3 cells we observed a concentration threshold for efficient EGFP expression. Very clearly, this transition requires an extension of the Leonhardt model. Either polyplexes act in a cooperative fashion inside endosomes or uptake first occurs by an endocytic mechanism that does not lead to endosomal release, and only once this process is close to saturation, endocytosis occurs by a mechanism that leads to endosomal release.

The stronger endosomal capture for PF14 polyplexes is consistent with the higher degree of lysosomal colocalization and also with the more pronounced loss of Cy5 fluorescence over time. For LipoMM, lysotracker fully colocalized with Cy5-mRNA, whereas Cy5-mRNA did not fully colocalize with lysotracker. This observation indicates that for LipoMM uptake also occurs by endocytosis, followed by a more efficient endosomal release, or that mRNA is delivered in part by direct fusion of particles with the plasma membrane.

Overall, our results show that mRNA provides excellent control over dosing. The maintenance of linearity over 72 h—in spite of differences in protein expression for the various time points but up to a factor of 100—demonstrates that the initial molecular processes are not subject to saturation. Even in vivo, dose and protein expression correlated linearly over 3 orders of magnitude. At this point, we restricted the in vivo analyses to LipoMM-formulated mRNA, which preferentially targets the spleen. It will be interesting to investigate the extent to which the same linearity holds for other organs and for mRNA delivered by other types of formulations. The high control over protein expression via mRNA dose is highly beneficial for future clinical applications, where, depending on the protein of interest, the required expression levels may differ widely to achieve the desired effect. Transcription factors, for example, require lower doses than structural proteins or metabolic enzymes. The excellent control of dose and effect for this novel class of therapeutics will render mRNA a finely tunable precision medicine.

## Figures and Tables

**Figure 1 biomedicines-09-00511-f001:**
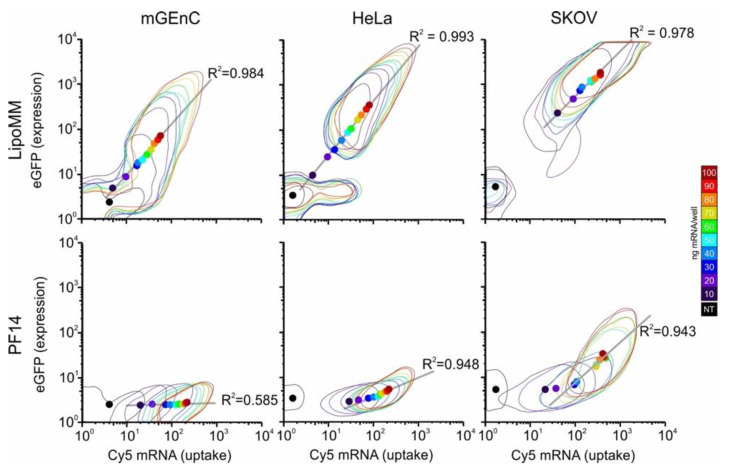
Protein expression shows a linear correlation with mRNA dose. Cells were incubated with the indicated doses in a total volume of 250 µL for 24 h, followed by flow cytometry. The 5% contours, as generated with flowJo software (containing 95% of cells) are displayed. The geometric mean of each dose is indicated by a dot colored in correspondence to the 5% contours. For PF14/Hela and PF14/SKOV-3, untreated cells and cells treated with the lowest two concentrations were excluded from the linear correlations, and only those cells that showed expression were included. For each cell type and transfection condition, 2–3 independent experiments were conducted. One representative experiment is shown.

**Figure 2 biomedicines-09-00511-f002:**
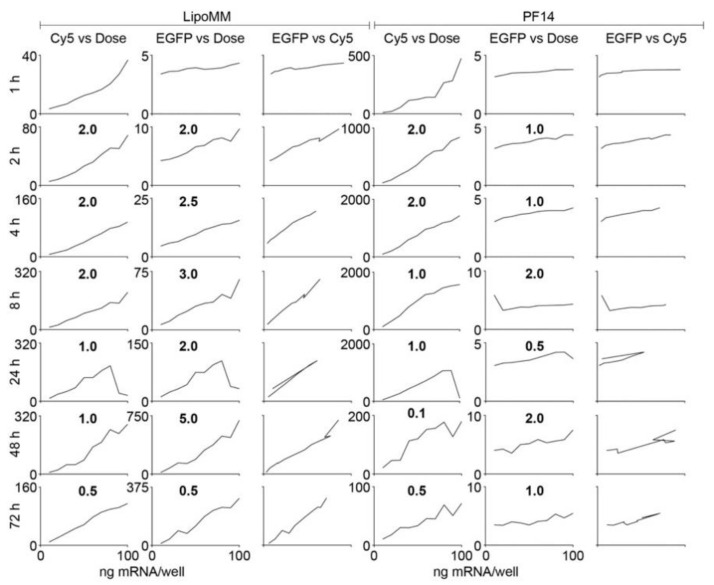
The linear dependencies of dose, uptake, and expression are maintained over time for mGEnC. The left three columns show uptake versus dose, expression versus dose, and expression versus uptake for LipoMM treated cells; the right three columns show the same set of graphs in the same order for PF14 treated cells. The *y* axes are scaled to enable the best representation of data. To stress the scaling differences, the factors by which the maximum values differ between consecutive panels are indicated in between the panels. For the EGFP versus Cy5 panels, the scaling corresponds to the *y* axis scaling of the EGFP versus dose and Cy5 versus dose panels. These data are derived from the primary data shown in Appendix A.

**Figure 3 biomedicines-09-00511-f003:**
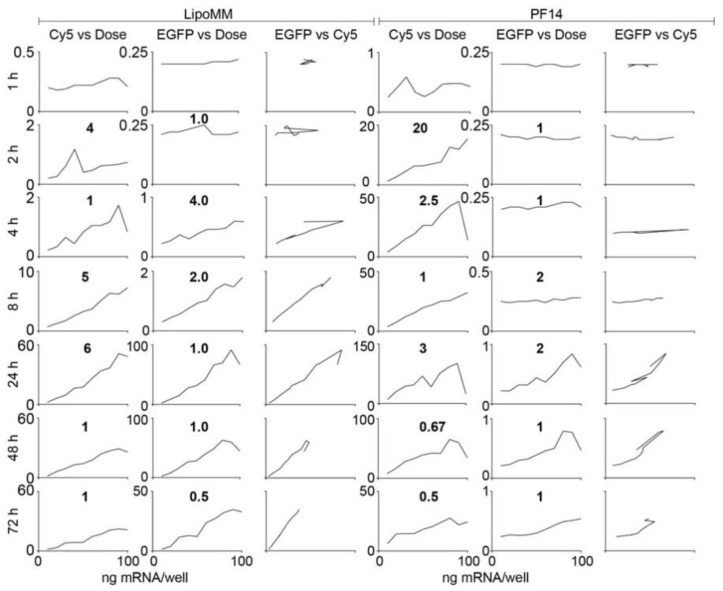
The linear dependencies of dose, uptake, and expression are maintained over time for HeLa cells. The left three columns show uptake versus dose, expression versus dose, and expression versus uptake for LipoMM treated cells; the right three columns show the same set of graphs in the same order for PF14 treated cells. The *y* axes are scaled to enable the best representation of data. To stress the differences in the *y* axis scaling, the factors by which the maximum values differ between consecutive panels are indicated in between the panels. For the EGFP versus Cy5 panels, the scaling corresponds to the *y* axis scaling of the EGFP versus dose and Cy5 versus dose panels. These data are derived from the primary data shown in Appendix A.

**Figure 4 biomedicines-09-00511-f004:**
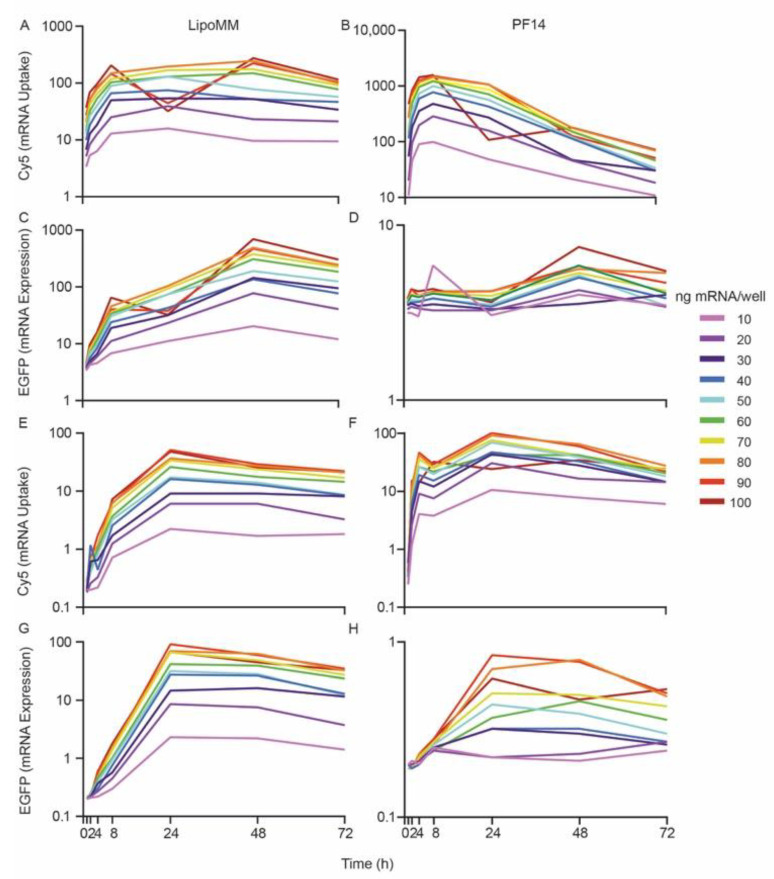
The temporal evolution of mRNA uptake and protein expression is largely independent of mRNA dose. Time courses of mRNA uptake and EGFP expression for mGEnC (**A**–**D**) and HeLa cells (**E**–**H**); left columns depict cells treated with LipoMM-formulated mRNA, while right columns depict cells treated with PF14-formulated mRNA. In order to render the initial increase in uptake and expression more readily discernible, *y* axes are plotted on a logarithmic scale. For linear scale refer to Appendix A. These data are derived from the primary data shown in Appendix A.

**Figure 5 biomedicines-09-00511-f005:**
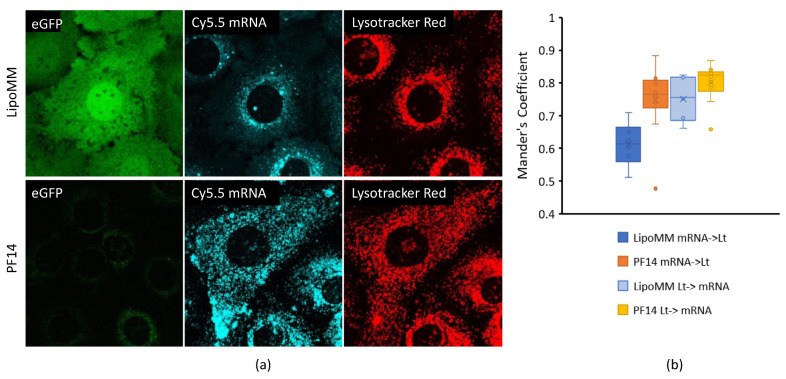
Peptide-mediated delivery leads to a stronger lysosomal retention of mRNA than lipid-mediated delivery. (**a**) mGEnC were incubated with 300 ng/well Cy5-mRNA coding for EGFP, using Lipofectamine MessengerMax (LipoMM) or PF14 for 24 h. Lysotracker Red was added 30 min prior to microscopy. Shown are data for one out of two independent experiments. (**b**) Colocalization was determined by calculating the Mander’s coefficient from at least 5 different fields per condition. Mean values are shown for the presence of Cy5-mRNA in lysotracker-positive pixels and the other way around.

**Figure 6 biomedicines-09-00511-f006:**
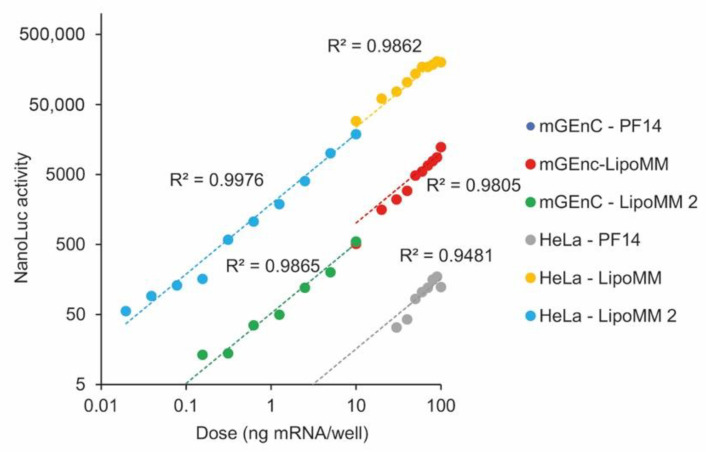
Reporter gene expression correlates linearly with mRNA dose independent of cell line and delivery vehicle. Hela cells and mGEnC were incubated with the indicated mRNA doses for 24 h, followed by harvesting of supernatants for determination of luciferase activity. Different concentration ranges were tested in independent experiments. For mGEnC transfected with PF14, no expression was detected. Shown are data for one out of 2–3 independent repetitions.

**Figure 7 biomedicines-09-00511-f007:**
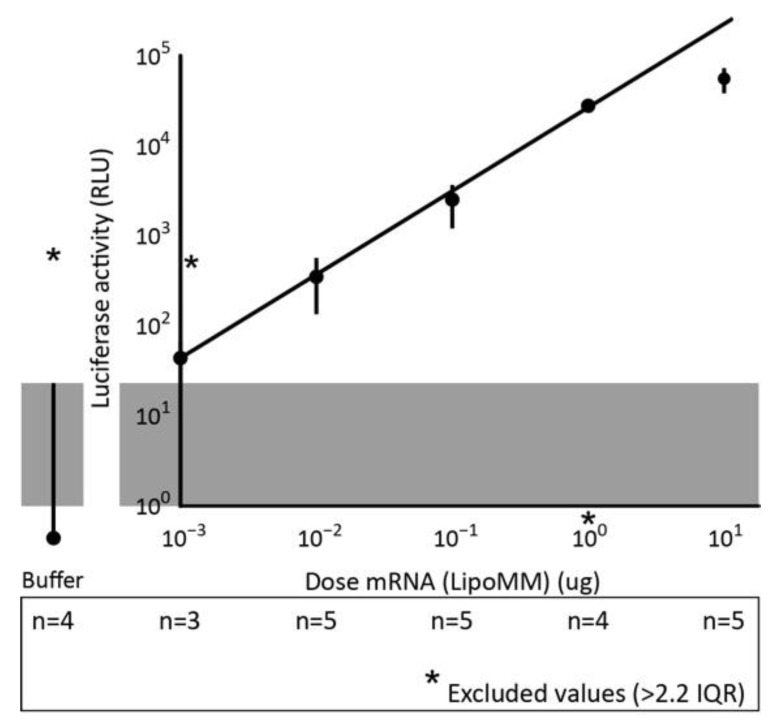
Additionally, in vivo, reporter gene expression linearly correlates with of mRNA dose. BALB/c mice were injected with the indicated dose of LipoMM-formulated firefly luciferase mRNA, and after 24 h luciferase activity was determined in spleen lysates. The average luciferase activity of 5 mice is shown.

**Figure 8 biomedicines-09-00511-f008:**
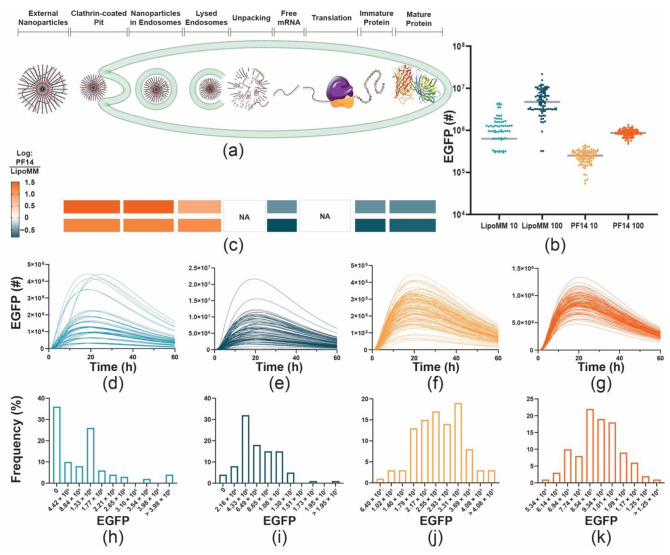
Differences in the number of mRNA molecules per nanoparticle suffice to explain differences in expression levels between PF14- and LipoMM-mediated delivery. Messenger RNA delivery and expression were simulated with a multistep kinetic model with the only difference being that for PF14, 3 mRNA molecules per nanoparticle were assumed, whereas for LipoMM 300 were used. (**a**) Schematic depiction and individual steps of the model. Steps for which ratios are given in (**c**) were explicitly considered by the model. Following uptake, the delivery agents accumulate in endosomes, of which a fraction will eventually lyse and release the delivery agents into the cytosol. After unpacking, the mRNA will be translated into immature (non-fluorescent) EGFP, followed by maturation into fluorescent EGFP that corresponds to the analytical readout. (**b**) Peak EGFP expression levels at 21 h after the beginning of the transfection. (**c**) Ratios of quantities for 10 and 100 ng mRNA delivered by PF14 and LipoMM. The ratios are reflected by the color code on the right. (**d**–**g**) Time course of cellular EGFP numbers for a total of 100 stochastic simulations each, for 10 ng and 100 ng mRNA delivered by LipoMM (**d**,**e**) and PF14 (**f**,**g**). (**h**–**k**) Frequency distributions of peak intracellular EGFP concentrations for 10 and 100 ng mRNA delivered by LipoMM (**h**,**i**) and by PF14 (**j**,**k**).

**Table 1 biomedicines-09-00511-t001:** A larger number of mRNA per nanoparticle suffices to explain the higher median EGFP expression for LipoMM and the larger fraction of expression-negative cells. Key input and output parameters of the kinetic model. Input parameters are given in the first two sections of the table, output parameters below.

	PF14 10 ng	LipoMM 10 ng	PF14 100 ng	LipoMM 100 ng
mRNA molecules	1.881 × 10^10^		1.881 × 10^11^	
mRNA/cell	1.881 × 10^5^		1.881 × 10^6^	
Particles/cell ^1^	62,700	627	627,000	6270
Endosomes/cell	65	6	208	20
Average EGFP/cell	2.5 × 10^5^	8.7 × 10^5^	8.5 × 10^5^	5.6 × 10^6^
Geometric mean EGFP/cell	2.33 × 10^5^	1.05 × 10^6^	8.59 × 10^5^	4.77 × 10^6^
Fraction of EGFP-negative cells	0%	37%	0%	5%
Ratio EGFP LipoMM/PF14	3.5		6.5	

^1^ To calculate the particles per cell, we considered a fully confluent well of an 8-well chambered coverslip that contained 1 × 10^5^ cells.

## Data Availability

Data are available for reuse upon reasonable request.

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
