# Peer review of "Protein Expression Correlates Linearly with mRNA Dose over Up to Five Orders of Magnitude In Vitro and In Vivo"

_biomedicines, 2021, doi:10.3390/biomedicines9050511_

Round 1
Reviewer 1 Report
In the paper “Protein expression correlates linearly with mRNA dose over up to five orders of magnitude in vitro and in vivo” The authors investigated the dose-response functions of lipofectamine and PF14 on the level of EGFP protein in the total cell population and on individual cells. The cells employed were glomerular endothelial cells (mGEnC), SKOV-3 ovarian cancer cells, and HeLa cells as cellular models, while for the in vivo study BALB/cJRj mice were used. The analysis and the presentation of the experimental results are accomplished satisfactorily however, few points require further consideration.
- Did the authors perform any Cytotoxicity assay on the cell lines? Why did the authors decide to use only one concentration? Please clarify.
- Please revised in material and methods the phrase “Pepfect14 …., a kind gift of Dr. Hällbrink”
- Please revised in the discussion, the unit of measurement in the phrase “Based on the number of polyplexes that were deposited on a glass surface, they postulated uptake of only 4 – 8 polyplexes per cell at an mRNA dose of 1 g in 1 ml total incubation volume.”
- I suggest to the authors to remove or improve the part relative to experiments on in vivo models. This part is really poorly planned and explained in material and methods. Moreover, this part both in the results and the discussion is not really correlated and supported with the in vitro data.
Author Response
We would like to thank the reviewer for the careful reading of our manuscript. Below, please find the point-by-point reply to the reviewer’s comments.
- Did the authors perform any Cytotoxicity assay on the cell lines? Why did the authors decide to use only one concentration? Please clarify.
Answer: In the context of this project, we have not explicitly performed toxicity experiments as we restricted ourselves to concentrations for which we had determined the absence of toxicity before. In the revised version we added (l. 207) “and both vectors were used at concentrations of which we had determined before that they did not cause any toxicity [14]“ with reference to this earlier work. We do not know what the reviewer refers to with the comment on use of only one concentration as the whole manuscript is about dose/response functions.
- Please revised in material and methods the phrase “Pepfect14 …., a kind gift of Dr. Hällbrink”
Answer: We changed the phrase to “Pepfect14 (Stearyl-AGYLLGKLLOOLAAAALOOLL-NH2, where O stands for ornithine and –NH2 for C-terminal amidation) was a kind gift of Dr. Hällbrink (Stockholm University, Sweden). The peptide was dissolved …”
- Please revised in the discussion, the unit of measurement in the phrase “Based on the number of polyplexes that were deposited on a glass surface, they postulated uptake of only 4 – 8 polyplexes per cell at an mRNA dose of 1 g in 1 ml total incubation volume.”
Answer: The unit was corrected to µg.
- I suggest to the authors to remove or improve the part relative to experiments on in vivo models. This part is really poorly planned and explained in material and methods. Moreover, this part both in the results and the discussion is not really correlated and supported with the in vitro data.
Answer: As we consider the in vivo experiments a valuable extension of the in vivo experiments we extended the explanation in the materials and methods section: “10 µg Fluc mRNA was formulated with Lipofectamine MessengerMax as described above, and diluted in Opti-MEM to the concentration used for injection, which took place within 30 minutes after formulation. BALB/cJRj mice (Janvier Labs, Le Genest-Saint-Isle, France) were injected in the tail vein with 50 µl of the indicated concentration of formulated mRNA.” Also, the explanation of results was adjusted to enhance clearity. In the discussion section we added: “At this point we restricted the in vivo analyses to LipoMM-formulated mRNA which preferentially targets the spleen. It will be interesting to investigate to which extent the same linearity holds for other organs and for mRNA delivered by other types of formulations.”
Reviewer 2 Report
The manuscript presents a detailed analysis of mRNA delivery using liposomal and cell penetrating peptide based formulations. The manuscript is well written and overall the manuscript meets the standards to be accepted for publication in Biomedicines. Here are a few comments that the authors could address before the manuscript is published
- What is the reason for the authors choosing liposomes and the cpp? please provide a justification for this approach.
- The authors used the cy5 intensity instead of uptake - It is highly recommended to have the terms directly stated - Uptake and expression (instead of Cy5 and EGFP etc)
- Can the authors explain why there are liposomal particles that have low mRNA loading - can there be a technique to track this statistically within a given sample?
Author Response
We would like to thank the reviewer for the careful reading of our manuscript. Below, please find the point-by-point reply to the reviewer’s comments.
- What is the reason for the authors choosing liposomes and the cpp? please provide a justification for this approach.
Answer: Lipofection, i. e. the non-covalent complexation with lipid-based transfection agents in THE standard for delivery of mRNA in the laboratory (ll. 53-55). We would like to point out, that lipofection is not equivalent to encapsulation in liposomes. As detailed in the introduction ll. 56 – 62, complexation with cationic cell-penetrating peptides is a widely explored approach for oligonucleotide delivery and in particular the peptide Pepfect 14 has shown activity before. Given the interest in both fields (CPPs and mRNA), investigating CPP-mediated delivery is therefore well justified.
- The authors used the cy5 intensity instead of uptake - It is highly recommended to have the terms directly stated - Uptake and expression (instead of Cy5 and EGFP etc)
Answer: We would prefer to stay with the axis labels and phrasing as is as Cy5 is the measure of uptake and the Cy5 and EGFP signals are actually measured. In fact, the axis labels explicitly state that Cy5 is the measure for mRNA uptake.
- Can the authors explain why there are liposomal particles that have low mRNA loading - can there be a technique to track this statistically within a given sample?
Answer: We assume that this must be a misunderstanding. We stated that because for LipoMM there are more mRNA molecules per particle, there is more heterogeneity in the number of nanoparticles that a given cell may receive. Publications in which individual particles per cell have been detected have been published (see for example: Nucleic Acids Research, 2017, doi: 10.1093/nar/gkx290)
Other changes:
We had noted that ref 23 had been cited erroneously instead of ref 14. Therefore ref. 23 was removed from the reference list.
Round 2
Reviewer 1 Report
I am satisfied with the updated manuscript provided by authors. Hence, I request to accept this manuscript in Biomedicines.